# The Association between Cardiorespiratory Fitness and Reported Physical Activity with Sleep Quality in Apparently Healthy Adults: A Cross-Sectional Study

**DOI:** 10.3390/ijerph18084263

**Published:** 2021-04-17

**Authors:** Ahmad M. Osailan, Ragab K. Elnaggar, Saud F. Alsubaie, Bader A. Alqahtani, Walid Kamal Abdelbasset

**Affiliations:** 1Physical Therapy and Health Rehabilitation Department, College of Applied Medical Sciences, Prince Sattam bin Abdulaziz University, Alkharj 16278, Saudi Arabia; r.elnaggar@psau.edu.sa (R.K.E.); s.alsubaie@psau.edu.sa (S.F.A.); Ba.alqahtani@psau.edu.sa (B.A.A.); w.kamal@psau.edu.sa (W.K.A.); 2Department of Physical Therapy for Pediatrics, Faculty of Physical Therapy, Cairo University, Giza 12613, Egypt; 3Department of Physical Therapy, Kasr Al-Aini Hospital, Cairo University, Giza 12613, Egypt

**Keywords:** cardiorespiratory fitness, VO_2PEAK_, sleep quality, physical activity

## Abstract

Background: Recently, poor cardiorespiratory fitness (CRF) has been postulated as an adverse health outcome related to poor sleep quality. However, studies investigating the relationship between CRF and a subjective sleep quality index are scarce. Thus, the current study aimed to investigate the association between CRF and the Pittsburgh Sleep Quality Index (PSQI) in apparently healthy people. The secondary aim was to investigate the association between reported physical activity (PA) and PSQI. Methods: Thirty-three healthy male participants volunteered to participate. CRF (VO_2PEAK_) was measured via cardiopulmonary exercise testing on a treadmill. A short form of the International Physical Activity Questionnaire (IPAQ) was used to measure PA, and PSQI was used for the sleep quality index. Results: There was no correlation between CRF and PSQI total score or any component of the PSQI. There was a significant inverse correlation between IPAQ and PSQI total score (r = −0.36, *p* = 0.04). Categorical data analysis of the two questionnaires revealed that 42.4% of the participants who reported low physical activity also had poor sleep quality. Conclusions: The current study showed no association between CRF and the subjective sleep quality index but demonstrated a moderate inverse association between reported PA and subjective sleep quality index. The findings suggest that the more reported PA, the better the overall sleep quality.

## 1. Introduction

Poor sleep quality is considered a public health burden and has been associated with many health problems [1]. Fatigue, tiredness, and daytime sleepiness have been reported by those who experience short sleep duration [2]. Poor sleep quality has also been linked with many adverse health outcomes, including hypertension [3], increased risk of diabetes [4], risk of cardiovascular disease (CVD) [5], and poor cardiorespiratory fitness (CRF) [6]. These health issues may eventually be attributed to the increased risk of mortality among people with poor or disturbed sleep quality [7].

CRF refers to the heart, lungs, and circulatory system’s ability to deliver oxygenated blood according to the metabolic demands required by the large group of muscles during heavy, dynamic activity [8]. CRF has been strongly associated with cardiovascular health. In this context, poor sleep quality has been suggested as a factor related to reduced CRF [9]. Having reduced CRF may consequently lead to an increased risk of CVD [10]. Mounting evidence suggests that regular exercise has a favorable effect on sleep disturbances [11]. However, the question about the relationship between CRF and sleep quality is debatable.

Studies investigating the relationship between CRF and sleep quality are scarce. In a longitudinal study, CRF was not associated with sleep disturbances among women diagnosed with stress-related exhaustion disorder [12]. The association between CRF and sleep quality was reported among apparently healthy adolescent girls [9]. In another longitudinal study, it was reported that middle-aged people tend to have lower CRF due to increased sleep complaints [13]. The inverse association between CRF (measured via VO_2PEAK_) and insomnia was also reported in [8]. Most of the previous studies measured CRF via estimation based on activity, except for the latter one.

Furthermore, previous studies have focused on populations with clinical conditions, and to the best of knowledge, limited studies have explored this relationship in healthy people using CRF (measured via VO_2PEAK_) and sleep quality. Given the lack of studies in this area among healthy people with a standardized exercise testing protocol, this study’s primary aim was to investigate the association between CRF and sleep quality in apparently healthy people. The secondary aim was to investigate the association between reported physical activity and sleep quality. It was hypothesized that CRF and reported physical activity would be inversely associated with sleep quality.

## 2. Materials and Methods

### 2.1. Participants

Thirty-three apparently healthy men (i.e., free from chronic disorders) were recruited from Prince Sattam bin Abdulaziz University to participate in the study from December 2019 to December 2020. Inclusion criteria were as follow: apparently healthy, above 18 years old. Exclusion criteria were: cardiovascular diseases, pulmonary diseases, or any other diseases that can limit lung capacity (e.g., cold, flu), history of pulmonary surgery, neurological diseases, having a recent musculoskeletal injury [14], and any comorbidity incompatible with exercise testing as per the American College of Sports Medicine (ACSM) [15]. Also, participants with a history of illness that limits physical exertion and recent surgery limiting physical work were excluded. According to the sample analysis performed using PASS software, V 14.0.15 (NCSS, Kaysville, UT, USA), a sample size of 32 subjects produces a two-sided 95% confidence interval with a width equal to 0.597 when the estimate of the Spearman’s rank correlation is 0.45, which was obtained from analyzing data from the first 10 observations for the CRF (represented by the VO_2PEAK_ values) and sleep quality as reported on the Pittsburgh Sleep Quality Index (PSQI) [16,17]. The study was conducted according to the guidelines of the Declaration of Helsinki and approved by the Ethical Committee of the Deanship of Scientific Research at Prince Sattam bin Abdulaziz University (RHPT/019/056). All participants signed informed consent before participation.

### 2.2. Protocol

A total of thirty-five participants were screened for the study (see Figure 1). Participants attended one visit to the exercise research lab at the College of Applied Medical Sciences at Prince Sattam bin Abdulaziz University. Once accepted to participate, a brief summary of the study’s protocol was given. Participants were instructed not to consume caffeinated drinks, eat or smoke at least 3 h before the visit as per standard for exercise testing and training [18]. Before exercise testing, participants were asked to fill out the PSQI and the short form of the International Physical Activity Questionnaire (IPAQ). After completing the questionnaires, height was measured to the nearest 0.5 cm using a stadiometer, and weight was measured using a weight analyzer (DETECTO, U.S.A.). Resting brachial blood pressure was taken using an electronic sphygmomanometer while seated in a chair with an armrest (Wollex Blood Pressure Monitor (ARM)/WXT-5902, Cigli Izmir, Turkey). Then, participants were fitted with a Polar heart rate monitor (Polar H7) to monitor the heart rate during the test and an appropriate size face mask to cover the nose and mouth to measure inspired and expired gases for analysis (see Figure 1). Participants were given three minutes to measure their resting heart rate and resting metabolic rate as indicated by volume of O_2_ consumption while seated, followed by the graded exercise test (GXT). At the end of the GXT, a recovery period was applied. All the tests were standardized to be performed during the day between 9:00 a.m. to 2:00 p.m. Two of the participants had their test terminated at the early stages of the test due to severe leg pain without signs of volitional exhaustion. Thus, the final number of participants included in the analysis was 33 (see Figure 1).

### 2.3. Graded Exercise Test (GXT) Protocol

Cardiopulmonary exercise testing (CPET) was conducted using COSMED Quark to analyze inspired and expired gases and was calibrated each day before testing. Before GXT, participants were instructed to minimize talking as much as possible unless asked by the examiner to answer questions during the test. The face mask was connected to a breath-by-breath gas analyzer, COSMED Quark CPET. GXT was performed on a treadmill (HP Cosmos Mercury, Nussdoerf-Traunstien, Germany) using an individualized incremental test modified according to participants’ physical abilities [19]. The test started at a speed of the participant’s preference (approximately 3.5–4 kph) and 0% inclination. Once the test started, three minutes were given to serve as a familiarization/warming up phase with speed gradually increased to the participant’s ability. At the start of the third minute of the familiarization phase, the participant was encouraged to reach maximum brisk walking speed. After the third minute, the incremental test started when a constant speed was achieved (brisk walking) and 1% inclination. Throughout the GXT, the speed remained constant, and inclination was increased progressively every minute by 1%. Inspired and expired gases were analyzed breath-by-breath using COSMED Quark CPET throughout the test and during the recovery period for a minimum of 5 min. If the participant reached volitional exhaustion and could no longer complete the test or any relative or absolute contraindication of ACSM criteria were met, the test was terminated [15]. Following the test’s termination, participants were seated on a chair with an armrest while heart rate, and inspired and expired gases were checked. Blood pressure was continuously monitored during the recovery period for a minimum of six minutes or until vital signs regained normal ranges.

### 2.4. Outcome Measures

#### 2.4.1. Sleep Quality

Sleep quality was assessed using PSQI. The questionnaire is valid, reliable and consists of ten main questions comprising of 19 self-rated subjective questions [20]. Each participant was given brief information about the questions and was provided with assistance to explain some of the questions when necessary. The answers to the questions generate seven component scores. These components include sleep quality, sleep latency, sleep duration, habitual sleep efficiency, sleep disturbances, sleep medication use, and daytime dysfunction. The scores of these questions were dichotomized into the seven main components, with a range of 0 to 3 per each component, and a maximum score of 21 and a minimum of 0 for the whole questionnaire. A total score of <5 indicates good overall sleep quality whereas a total score ≥ of 5 indicates poor sleep quality [21].

#### 2.4.2. Physical Activity

Physical activity was assessed using the short form of the IPAQ, which is known to be valid and mainly consists of seven questions [22]. The participants were instructed to answer questions in the form based on their level of physical activity in the last seven days before testing. The brief information about the questionnaire (available as an introduction in the first sentences of the questionnaires) was read to the participants. Full attention was given to the participant to explain any further questions. The participant was given a pen to fill out the questionnaire in a paper format. The self-reported Physical Activity Questionnaire collects information about the duration and frequency of vigorous intensity, moderate intensity, and light intensity activities performed in the last seven days. After filling the questionnaire in a paper format, the answers were then copied to the IPAQ questionnaire’s Excel sheet. When the answers were copied to the Excel sheet, the participant’s age and weight were also entered. An overall estimation of metabolic equivalents (MET) (1 METs = 3.5 mL/kg/min of VO_2_) of physical activity performed was conducted using the automatic report provided from (http://www.ipaq.ki.se, accessed on 27 May 2019). The final report provided an estimation of the sum of MET per week. 

#### 2.4.3. Cardiorespiratory Fitness (VO_2PEAK_)

Peak oxygen uptake was recorded during treadmill GXT using a breath-by-breath gas analyzer. The inspired and expired gases data were collected and standardized to be averaged every five seconds. To minimize fluctuation of the data, final data collected were averaged for every 30 s of the VO_2_ sampling rate. VO_2PEAK_ was defined as the highest VO_2_ recorded during the GXT and was expressed as VO_2 mL_/kg/min. Achieving respiratory exchange (RER) ratio ≥ 1.1 was one of the main criteria to assess VO_2PEAK_ and ensure that the participants exercised to the maximum ability. Other additional criteria were achievement of heart rate ≥ 90% of age-predicted maximum heart rate. The majority of the participants in the current study achieved RER ≥ 1.1 and had other reasons apart from increased heart rate to stop the test and record Peak oxygen uptake, such as reaching maximum exhaustion indicated by having ≥ 8 on a 0–10 scale of perceived exertion or reporting severe leg pain/cramps during GXT.

### 2.5. Statistical Analysis 

Statistical analysis was performed using the Statistical Package for Social Sciences (IBM SPSS) (version 27, Armonk, NY, USA). The normality of the variables was tested using the Kolmogorov–Smirnov test. Normally distributed variables were presented as the mean and standard deviation and non-normally distributed variables as median and interquartile range. To investigate if smoking has any influence on the study’s main outcome measures, an independent *t*-test was used to compare the main outcome measures between smokers and non-smokers. Point biserial correlation was used to investigate the relationship between continuous data and categorical data (e.g., cardiorespiratory fitness and smoking). Bivariate correlation using Spearman’s rank correlation coefficient analysis was used to assess the relationship between cardiorespiratory fitness, reported physical activity and sleep quality. Cross tabs were also used to investigate the frequency and contingency between the categorical data of reported physical activity (IPAQ) and sleep quality (PSQI). The level of significance was set at *p* ≤ 0.05.

## 3. Results

The demographic characteristics of the participants are presented in Table 1. Some of the participants were smokers, but there was no difference in any of the main outcome measures between the smokers and non-smokers (see Table 2). Additionally, no significant correlation was found between smoking and CRF in the current sample (r(31) = −0.23, *p* = 0.19). Many of the participants had a BMI above normal. An inverse association was found between CRF and BMI (r(31) = −0.63, *p* < 0.001). However, this association did not influence the relationship between CRF and PSQI when BMI was entered as a confounding variable. The average sleep quality was poor, as indicated by the participants’ mean PSQI [21]. The reasons for exercise testing termination are shown in Table 1. A flow diagram of the study is presented in Figure 1.

### 3.1. Correlation between VO_2PEAK_ and PSQI and Its Components

Correlational analysis was conducted to explore the association between VO_2PEAK_ and PSQI with all its components. There was no significant association between VO_2PEAK_ and the total score of the PSQI or any of the components of the sleep quality index (see Table 3). No association was also found between CRF and total PSQI score when BMI was entered as a confounding variable (r(30) = −0.14, *p* = 0.43). As a sub-analysis, the correlation between IPAQ and PSQI total score was investigated. Table 4 showed a moderately significant inverse association between IPAQ and PSQI (r(31) = −0.36, *p* = 0.04). When all the components of sleep quality were investigated for correlational analysis, there was also an inverse significant moderate association between IPAQ and sleep medication (r (31) = −0.4, *p* = 0.02) (see Table 4).

### 3.2. Cross-Tabulation of the Categorical Data between IPAQ and PSQI

Cross tabs were used to investigate the frequency and the contingency between the categorical data of IPAQ and PSQI. IPAQ was categorized into three subgroups (low, moderate, high physical activity) based on the automatic report generated from the Excel sheet (provided by the manufacturer). The frequency of good sleep quality and poor sleep quality based on the categorization of reported physical activity is demonstrated in Table 5. Of the total sample, 84.8% of the participants had poor sleep quality (PSQI ≥ 5). Among participants with low PA (*n* = 15), 93.3% of participants had poor sleep quality. A total of 42.4% of the participants reported low physical activity and had poor sleep quality (see Figure 2). The number of participants with poor sleep quality decreased in the higher categories of PA.

## 4. Discussion

The current study explored the relationship between CRF (measured via VO_2PEAK_), self-reported IPAQ (short form) and sleep quality index score (measured via PSQI). Contrary to our hypothesis, the study showed no significant relationship between CRF and PSQI, but there was a moderately significant inverse relationship between IPAQ and PSQI. This indicates that the more people stay active, the better their overall sleep quality. The study also showed that with increased reported physical activity on IPAQ, fewer participants had poor sleep scores on the PSQI.

Cardiopulmonary exercise testing is well-known as a gold standard for the measurement of CRF. CRF refers to the cardiopulmonary system’s ability to transport oxygen from the atmosphere to the mitochondria to contribute to physical work performance. It also represents a quantification of functional capacity, which relies on multiple systems to meet the physical work’s metabolic demands. Thus, it is considered a reflection of total body health [23]. CRF can be measured directly or via estimation from the peak work achieved during physical work. Measurement of CRF via direct measurement of volumes of oxygen during physical work (e.g., breath analysis) is considered more objective and precise. In the current study, the use of direct measurement using breath-by-breath analysis was preferred. In this context, the study showed that the average VO_2PEAK_ of the participants was low, indicating reduced CRF, which is alarming, considering the participants’ age. This is possibly due to the lack of PA engagement, even for participants in this age range.

One of the other factors that could influence CRF is excessive weight gain and high fat percentage [24]. Although BMI is not an accurate marker for fat-related weight gain, it is still a common measure for healthy weight. In the current study, more than half of the participants had a BMI above normal (≥24.9 kg/m^2^), which may contribute partially to the reduced CRF in our participants (see Table 1). An inverse association was observed between CRF and BMI in the current sample. However, when the association between CRF and PSQI was investigated with BMI as a confounding variable, there was still no significant relationship between CRF and PSQI.

Limited studies have investigated the association between CRF and other parameters related to the quality of sleep. A longitudinal study reported no association between CRF and sleep disturbances in (*n* = 88) females diagnosed with stress-related exhaustive disorders [12]. On the contrary, a cross-sectional study reported that poor CRF was weakly associated with poor sleep quality among (*n* = 552) adolescent girls [9]. In the same context, a longitudinal epidemiological study found that a decline in CRF accelerates the risks of sleep problems, especially in middle-aged people [13]. The studies above utilized CRF estimation from peak physical work, and different methodologies were used to assess the association between CRF and sleep quality and different scales and outcome measures for sleep quality.

Few other studies have utilized direct measurement of CRF and its association with sleep problems. A study consisting of apparently healthy men and women (*n* = 3489) reported a moderate association between CRF (measured via VO_2PEAK_) and sleep problems (known as insomnia) [8]. Similar to our study’s findings, a cross-sectional study reported no association between CRF and PSQI among (*n* = 28) healthy males [25]. On the contrary, a recent cross-sectional study in sedentary middle-aged people (*n* = 74) reported better CRF (measured via VO_2MAX_) was related to better PSQI scores [26]. The relationship between CRF and sleep quality (PSQI) existed when the participants were older people as shown in the latter study [26]; however, when the studied population were younger, as in the former study [25] and the current study, no relationship was found. Despite some differences in methodology, the former two studies by Antunes et al. [25] and Mochón-Benguigui et al. [26], to some extent, used similar outcome measures as those used in the current study, but with different age categories (see Table 6). This may indicate that the adverse outcomes of poor sleep quality, such as poor/reduced CRF, are more evident among older people and less prominent at a younger age. Indeed, age was considered as one factor contributing to variation in the results reported in a systematic review on the inter-relationship between sleep and exercise [11]. Furthermore, the relationship between CRF and sleep in both older and young adults was investigated. It was found that the relationship existed among older people but was absent among young adults [27].

It must be noted that the CRF of the participants in the current study is lower than previous similar studies. The mean VO_2PEAK_ for our participants was 28.4 (mL/kg/min), which is lower than that reported by Antunes et al. [25] and Mochón-Benguigui et al. [26] (53 mL/kg/min, and 30.5 mL/kg/min, respectively). However, it must be noted that both previous studies used VO_2MAX_, which requires attainment of a plateau in maximum oxygen uptake, unlike VO_2PEAK_, which corresponds to the attainment of maximum oxygen uptake without plateau. Additionally, one of the inclusion criteria in a previous study was BMI ≤ 25 kg/m^2^ [25], which indicates some bias toward healthy people with less BMI. This might have contributed to obtaining higher VO_2_ than the current study’s participants. Despite the difference in the participants’ age, the value of the VO_2_ achieved during the exercise testing in the current study and the Mochón-Benguigui et al. study is relatively close, and this might be due to using similar exercise testing protocols on a treadmill. However, Antunes et al. conducted the test using a cycle ergometer. It is well-known that the obtained physiological responses differ between a treadmill and cycle ergometer [28]. Supposedly, the volumes of oxygen uptake during exercise on a treadmill tend to be higher than what is obtained on the cycle ergometer. However, due to the variation in multiple factors, including the participants’ age, BMI, and PA level, direct comparison between the mentioned studies in regard to the volume of oxygen uptake is complicated.

As a secondary aim of the current study, we sought to investigate the association between reported physical activity and sleep quality. The study showed a moderate inverse association between reported physical activity and PSQI, meaning that the more people report being physically active, the better their overall sleep quality. In addition, it was found that a high percentage of people who reported low PA also reported poor sleep quality on the PSQI (see Figure 2). This may indicate that people with less PA are more likely to have overall poor sleep quality. It was suggested that there is a bidirectional association between PA and sleep quality through physiological and psychological mechanisms [29]. These physiological mechanisms include the regular benefits of PA, such as improvement in vagal regulation of the heart and in the endocrine system, whereas psychological mechanisms include improvement in mood and production of melatonin. These mechanisms contribute to better sleep, which eventually contributes to better and more engagement in PA [29]. However, in terms of association, the absence of an association between PA and sleep quality has been concluded by some studies when both PA and sleep quality were objectively measured [30], and also when PA was objectively measured and sleep quality was subjectively reported [31].

The inverse association between reported PA (IPAQ) and sleep quality using the PSQI has been reported in some studies. In parallel to the current study findings, an inverse association was reported between the total duration of PA and PSQI total score among aged people [26] and young participants [32]. It is worth noting that in the current study, the energy cost of PA (MET-min/week) for the reported PA was utilized, as it gives a more commonly used representation of the energy cost of activity in metabolic equivalents (1 MET = 3.5 mL/kg/min of VO_2_). However, in the previous studies, a different methodology was used. For example, one of the studies used reported physical activity in units of time [26], whereas the other study utilized the categorization of IPAQ as sufficient vs. insufficient PA [32]. This made a direct comparison between the previous studies’ results and the current one difficult due to different methodological approaches. The current study adds to the literature in that the PA’s energy cost (using IPAQ) for the reported PA was associated inversely with the total score of the PSQI index, which means that higher energy consumption in MET was related to better overall sleep quality.

Almost half of the participants in the current study reported low PA. This was not surprising considering the average VO_2PEAK_ of the participants (see Table 1), indicating a low CRF level. The prevalence of low PA in the current study is consistent with previous reports in the same country [33,34,35]. This is raising concerns about the future risk of comorbidities and cardiovascular diseases. Regarding the percentage of the participant with poor sleep, 85% experienced poor sleep, which is again consistent with previous studies [36]. This needs to be addressed in clinical practice and research by implementing more strategies to encourage further engagement in exercise and PA and more education about efficient sleep.

Despite the absence of association between CRF and sleep quality, the current study adds to the literature by providing further knowledge about this association among young adult groups using CRF, which was measured directly. This may warrant further research using more objective parameters for sleep quality, such as Actigraph or accelerometers, which would provide more robust data about the quantity and quality of sleep.

There are several limitations to the current study. Despite the good applicability and reproducibility of the PSQI, it has some limitations as some of its components are weakly associated with the total score [37]. Thus, future research may consider implementing objective tools to assess sleep quantity and quality, which may provide better information and better representation of sleeping habits. This would also allow better and more accurate detection of the relationship between sleep quality and CRF, especially when measured directly. Furthermore, in the current study, a subjective PA questionnaire was used via IPAQ, which does not replace the use of objective tools such as the accelerometer for PA measurement. Adding an objective measure for PA would also give more precise data on the participants’ level of PA. Another limitation is the small sample, and only males were included in the current study, limiting the generalizability of the results to the included participants only. Future research should be conducted on a larger sample with the inclusion of females. It must be noted that limitation in the number of participants happened due to the pandemic crisis, which made the recruitment of more participants complicated.

## 5. Conclusions

In conclusion, the current study found no association between CRF and sleep quality. However, there was an inverse association between reported PA and overall sleep quality. Additionally, a high percentage of people reported low PA and also had overall poor sleep quality. This may indicate that better sleep quality is associated with more engagement in PA. Future cross-sectional studies should include objective measures for the quality and quantity of sleep and investigate their association with direct CRF measurement. Future prospective studies should also investigate if more engagement in physical activity improves the quality of sleep.

## Figures and Tables

**Figure 1 ijerph-18-04263-f001:**
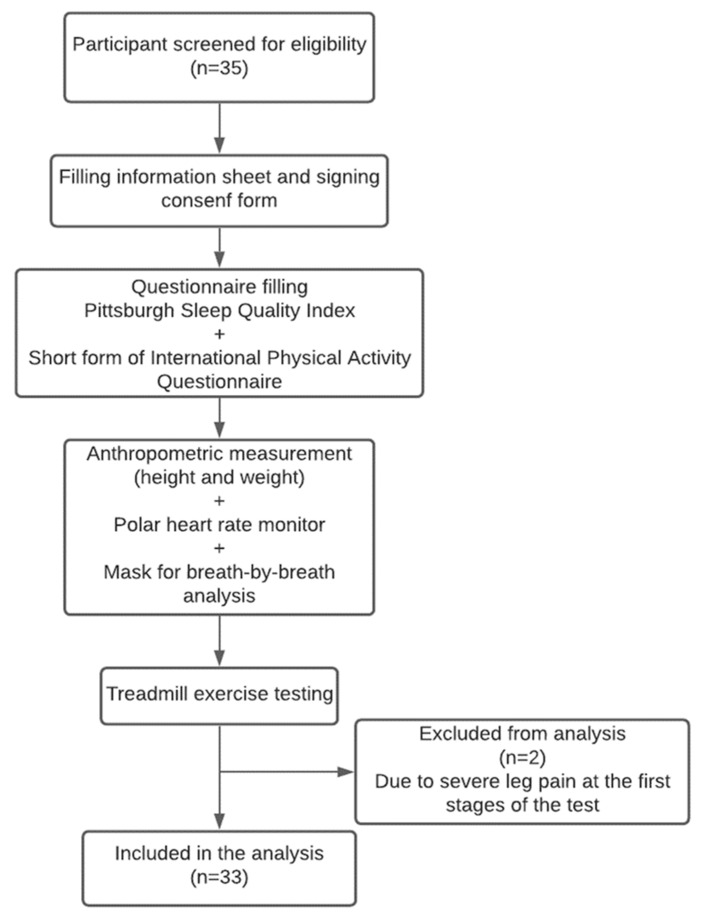
Flow diagram of the study protocol.

**Figure 2 ijerph-18-04263-f002:**
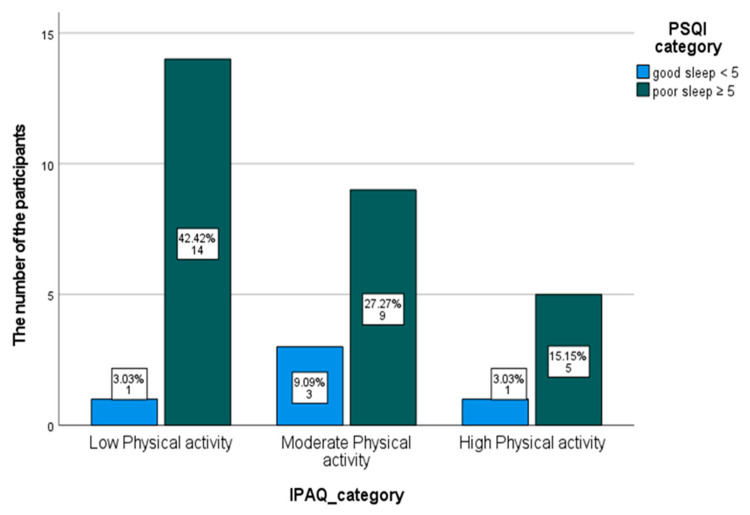
A bar chart of the number and frequency of IPAQ categories and PSQI categories.

**Table 1 ijerph-18-04263-t001:** Demographic characteristics of the participants.

Characteristic	Value
Age (years)	23 (22–24)
Weight (kg)	81.9 ± 17.9
Height (m)	1.72 ± 0.07
BMI (kg/m^2^)	27.8 ± 8.7
- Underweight (*n*, %)	1, 3%
- Normal weight (*n*, %)	10, 30.3%
- Overweight (*n*, %)	12, 36.4%
- Obesity class I (*n*, %)	7, 21.2%
- Obesity class II (*n*, %)	1, 3%
- Obesity class III (*n*, %)	2, 6.1%
Smoker (*n*, %)	10, 30.3%
**Sleep and physical activity variables**	
PSQI total score	7.3 ± 3.2
IPAQ (MET-min/week)	1173 (241–2203.5)
**Hemodynamic variables**	
Resting HR (bpm)	77 (70–80)
Resting SBP (mmHg)	127 ± 9.9
Resting DBP (mmHg)	83 (63–83)
HR maximum (bpm)	169 ± 13.6
HRR1	34.2 ± 12.8
Post-exercise test SBP	153 ± 14.5
Post-exercise test DBP	88 ± 8.01
RF (b/min)	39.4 ± 7.14
P_ET_CO_2_-Resting (mmHg)	33.3 ± 3.43
P_ET_CO_2_-Peak (mmHg)	44.5 ± 3.38
RER	1.15 ± 0.10
Vt peak (litre)	2.09 ± 0.45
VE peak (L/min)	74.04 ± 18.12
VO_2_ (L/min)	2.28 ± 0.48
VO_2PEAK_ (ml/kg/min)	28.4 ± 5.8
**Reasons for test termination**	
Maximum exhaustion (*n*, %)	11, 33.3%
Lower limb fatigue (*n*, %)	13, 39.4%
Request to stop (*n*, %)	2, 6.1%
SOB and HR higher than 90% HRr (*n*, %)	2, 6.1%
HR higher than 90% of HRr (*n*, %)	2, 6.1%
Severe SOB	3, 9.1%

Values are presented as mean and standard deviation or median (25th to 75th percentile) as appropriate. BMI; body mass index, PSQI; Pittsburgh Sleep Quality Index, IPAQ; International Physical Activity Questionnaire (short form), MET; metabolic equivalents. Underweight (below 18.5 kg/m^2^), normal weight (18.5–24.9 kg/m^2^), Overweight (25.0–29.9 kg/m^2^), Obesity class I (30.0–34.9 kg/m^2^), Obesity class II (35–39.9 kg/m^2^), Obesity class III (above 40 kg/m^2^), HR; heart rate, bpm; beat per minute; SBP; systolic blood pressure, DBP; diastolic blood pressure, HRR1, Heart rate recovery (peak heart rate–heart rate at 1 min post-peak heart rate) RF; respiratory frequency, P_ET_CO_2_; end-tidal carbon dioxide, RER; respiratory exchange ratio, Vt; tidal volume, VE; minute ventilation, SOB; shortness of breath, HRr; heart rate reserve (age-predicted maximum heart rate).

**Table 2 ijerph-18-04263-t002:** Comparison of the main outcome measures between smokers and non-smokers.

Variable	Smokers *n* = 10	Non-Smokers *n* = 23	*p*
PSQI	8 ± 2.4	7 ± 3.5	0.42
IPAQ (MET-min/week)	1251.6 ± 1768.4	2503 ± 4999.3	0.45
VO_2PEAK_ (ml/kg/min)	26.3 ± 5.9	29.2 ± 5.7	0.19

IPAQ; International Physical Activity Questionnaires, PSQI; Pittsburgh Sleep Quality Index.

**Table 3 ijerph-18-04263-t003:** Correlation analysis between VO_2PEAK_ and sleep quality index (PSQI).

Variable	VO_2PEAK_	
	*r*	*p*
Sleep quality	−0.21	0.91
Sleep latency	0.50	0.78
Sleep duration	0.60	0.73
Sleep efficiency	−0.15	0.40
Sleep disturbances	−0.29	0.09
Sleep medication	0.54	0.77
Sleep dysfunction	0.56	0.76
PSQI total score	−0.07	0.71

PSQI; Pittsburgh Sleep Quality Index.

**Table 4 ijerph-18-04263-t004:** Correlation analysis between IPAQ and sleep quality index.

Variable	IPAQ	
	*r*	*p*
Sleep quality	−0.18	0.31
Sleep latency	0.03	0.89
Sleep duration	−0.28	0.12
Sleep efficiency	−0.14	0.43
Sleep disturbances	−0.33	0.06
Sleep medication	−0.40	0.02
Sleep dysfunction	−0.18	0.33
PSQI total score	−0.36	0.04

IPAQ; International Physical Activity Questionnaires, PSQI; Pittsburgh Sleep Quality Index.

**Table 5 ijerph-18-04263-t005:** Frequency distribution between IPAQ and PSQI.

IPAQ Category	PSQI Category		Total
	Good Sleep Quality	Poor Sleep Quality	
Low PA *n* (%)	1 (6.7%) ^a^	14 (93.3%) ^b^	15 (45.5%) ^c^
Moderate PA *n* (%)	3 (25%) ^a^	9 (75%) ^b^	12 (33.3%) ^c^
High PA *n* (%)	1 (16.7%) ^a^	5 (83.3%) ^b^	6 (18.2%) ^c^
Total	5 (15.2%) ^a^	28 (84.8%) ^b^	33

IPAQ; International Physical Activity Questionnaire, PSQI; Pittsburgh Sleep Quality Index, PA; physical activity. ^a^; Indicate the percentage of participants with good sleep quality based on the IPAQ category’s total number. ^b^; Indicate the percentage of participants with poor sleep quality based on the IPAQ category’s total number. ^c^; indicates the percentage of participants in each category of IPAQ from the total sample.

**Table 6 ijerph-18-04263-t006:** A comparison between the current study and other studies using similar outcome measures.

Authors and Year	Participants’ Age (Mean or Median)	Body Mass Index Inclusion Criteria	Physical Activity Inclusion Criteria	Graded Exercise Testing	Assessment of Sleep Quality	Assessment of Physical Activity	Cardiorespiratory Fitness Variable	VO_2MAX_ Criteria	Results
Antunes et al. [25] (2017)	29 years	BMI ≤ 25 kg/m^2^	No restrictions	Using cycle ergometer	PSQI	Using IPAQ	VO_2MAX_	Not fulfilled or mentioned	No association between VO_2MAX_ and sleep quality
Mochón-Benguigui et al. [26] (2021)	53.7 years	BMI between 18.5 and 35 kg/m^2^	Sedentary	Using treadmill	Accelerometer + PSQI	Accelerometer	VO_2MAX_	Fulfilled	Inverse association between VO_2MAX_ and sleep quality
Current study	23 years	No restrictions	No restrictions	Using treadmill	PSQI	Using IPAQ	VO_2PEAK_	NA	No association between VO_2PEAK_ and sleep quality

IPAQ; International Physical Activity Questionnaires, PSQI; Pittsburgh Sleep Quality Index, BMI; body mass index, NA; not applicable.

## Data Availability

The data presented in this study are available on request from the corresponding author. The data are not publicly available due to the information contained that could compromise the privacy of research participants.

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
