# Peer review of "The Association between Cardiorespiratory Fitness and Reported Physical Activity with Sleep Quality in Apparently Healthy Adults: A Cross-Sectional Study"

_ijerph, 2021, doi:10.3390/ijerph18084263_

Round 1

Reviewer 1 Report

The study aimed to investigate the association between CRF and Pittsburgh sleep quality index (PSQI) in apparently healthy people, and the association between reported physical activities (PA) and PSQI.

In table 1, values are presented as mean and standard deviation, or median (25th to 75th percentile) as appropriate- results should be presented more clearly and BMI in WHO criteria.

In table 2, correlational analysis was conducted to explore the association between VO2 peak and PSQI with all its components- In general, if the authors expect a relationship between analyzed variables then I would expected analyses that go beyond simple correlations to investigate these relationships (e.g., multiply regression with variable selection, forrward or backward).

In table 3, the cross-tabulation was presented between categorical data (IPAQ vs PSQI), but I also suggest that you should analyze the relationship between the PSQI category and the values ​​of physical activity in MET.

 The study showed no significant relationship between CRF and PSQI. It is not clear why the VO2max scores of healthy young people are so low. The VO2 max results should also be presented in absolute values. The recorded average maximum frequency of heart rates indicates that the subjects may not have achieved the maximum effort. The authors should explain this problem further. In table 1, authors should present the recorded values ​​of: ventilation- Ve, respiratory frequency Rf, tidal volume Vt, oxygen pulse V02/HR. In discussion section, please compare the results of direct V02max measurements with the results of other authors.

Please explain more details in the manuscript: how the IPAQ was conducted (e.g. paper and pencil) and who and how instructed the subjects (procedure).

Author Response

Thanks to the reviewer for constructive feedback and comments. All responses and changes were highlighted in yellow in the manuscript for easy tracking. 

In table 1, values are presented as mean and standard deviation, or median (25th to 75th percentile) as appropriate- results should be presented more clearly and BMI in WHO criteria. 

Response

Thanks for this comment. The table has been amended to clearly classify the BMI according to WHO criteria. A sub-category of the variable was created to add more clearance. 

In table 2, correlational analysis was conducted to explore the association between VO2 peak and PSQI with all its components- In general, if the authors expect a relationship between analyzed variables then I would expected analyses that go beyond simple correlations to investigate these relationships (e.g., multiply regression with variable selection, forward or backward).

Response

Thanks for this comment. Indeed, it was our intention from the start to conduct a further analysis such as linear regression if there was an association between Cardiorespiratory fitness and the components of sleep quality. However, when no association was found, we did not conduct further assessment. The reason why we did not do this is because multivariate linear regression assumes that there is a linear relationship, and in the absence of a relationship between the variable, this was not possible. Furthermore, normality of variables is necessary to conduct a multivariate linear regression, which is violated with the components of sleep quality (according to Kolmogorov Smirnov test) as all were non-normally distributed, which is why we used Spearman rank correlation analysis only. 

In table 3, the cross-tabulation was presented between categorical data (IPAQ vs PSQI), but I also suggest that you should analyze the relationship between the PSQI category and the values ​​of physical activity in MET.

Response

Thanks for this comment. A further analysis was made to investigate the association between physical activity in MET and sleep quality index which showed no association with other components except for sleep medication. 

The study showed no significant relationship between CRF and PSQI. It is not clear why the VO2max scores of healthy young people are so low. The VO2 max results should also be presented in absolute values. The recorded average maximum frequency of heart rates indicates that the subjects may not have achieved the maximum effort. The authors should explain this problem further. In table 1, authors should present the recorded values ​​of: ventilation- Ve, respiratory frequency Rf, tidal volume Vt, oxygen pulse V02/HR. In discussion section, please compare the results of direct V02max measurements with the results of other authors.

Response

it is indeed not clear why CRF is low but we can speculate that the majority of the participants had a sedentary lifestyle as indicated by the IPAQ. As we mention in the manuscript, there is a great concern in the community that recommendations of physical activity are not met in the country.. We believe that the participants exercised tested to their maximum abilities as indicated by the average Respiratory exchange ratio of 1.15 ± 0.1. The CRF is low probably due to lack of participation and engagement of exercises even for the younger population. 

The VO2 peak in absolute value was added to table 1 as well as RF, tidal volume, and minute ventilation. However, adding VO2/HR was not possible to be added because HR was monitored via Polar HR and was not synchronized with data of breath-by-breath analysis. It would have possible if we conducted the tests with HR monitoring using ECG. 

With regards to the achieved maximum HR for the participant, the reason for test termination was not exclusive to reaching maximum HR, but complaints of exhaustion and fatigue were one of the common reasons. As well as other reasons as indicated by ACSM for termination of exercise testing. A list of reported reasons to terminate the test is added in the new version of the manuscript). 

With regards to the comparison with other studies about VO2 max, a paragraph was added in the discussion and highlighted in yellow. 

Please explain more details in the manuscript: how the IPAQ was conducted (e.g. paper and pencil) and who and how instructed the subjects (procedure).

Response 

Thanks for the comment. More details about how IPAQ was conducted was added where we explained that we used paper format and then the answers were copied to an excel sheet provided by the manufacturer. 

Reviewer 2 Report

Please add in exclusion criteria the recent injury (The Reciprocal Association between Fitness Indicators and Sleep Quality in the Context of Recent Sport Injury. doi: 10.3390/ijerph17134810)

The flow chart is unclear. Please replace the flow chart and added more information

The 30,3% of participants are smoker. Please add the pack years and the values of spirometry.

Please add in table 1 the values of arterial blood pressure in maximal effort and at the 1st minute of recovery (HR, BP, VO2) to exclude the cardiological diseases. Several of participants are smoker and obese

Please add in table 1 the values of arterial blood pressure in maximal effort and at the 1st minute of recovery (HR, BP, VO2) to exclude the cardiological diseases. Several of participants are smoker and obese.

Please refer in cause of each trial was terminated i.e. dyspnea, leg fatigue...etc

Please add in table 1 the ratio of VE/MVV.

Please add the value of resting and maximal effort of PETCO2. The COSMED Quark record the end-tidal volume of CO2. The increased values of CO2 in resting relate to sleep deprivation.

(The use of cardiopulmonary exercise testing in identifying the presence of obstructive sleep apnea syndrome in patients with compatible symptomatology. doi: 10.1016/j.resp.2019.01.010.

The effect of physical strain on breeders patients with obstructive sleep apnea syndrome. doi: 10.1016/j.resp.2018.11.009

Effects of sleep deprivation on cardiorespiratory functions of the runners and volleyball players during rest and exercise. doi:10.1556/APhysiol.96.2009.1.3

End-tidal pressure of CO2 and exercise performance in healthy subjects. doi:10.1007/s00421-008-0773-z

Effects of sleep deprivation on cognitive and physical performance in university students. doi: 10.1007/s41105-017-0099-5)

Author Response

Thanks to the reviewer for constructive and helpful feedback. All the comments were addressed and changes and amendments suggested by the reviewer are highlighted in turquoise in the main manuscript. 

Please add in exclusion criteria the recent injury (The Reciprocal Association between Fitness Indicators and Sleep Quality in the Context of Recent Sport Injury. doi: 10.3390/ijerph17134810)

Response

Very good point. Recent musculoskeletal injury was added to the exclusion criteria. 

The flow chart is unclear. Please replace the flow chart and added more information

Response

Thanks for the feedback. The flow chart has been replaced and further information about the number of total participants screened for the study was added and the reason why they were not included in the analysis. 

The 30,3% of participants are smoker. Please add the pack years and the values of spirometry.

Response

Thanks for the comment. Unfortunately, we don't have information about the number of cigarettes smoked per day or the years of smoking. When we asked the participants, it was a yes/no question. We believe that this would have added more information about the participants, and this will be our aim in future research to collect more information about those who smoke. Nevertheless, an analysis was conducted to examine if there is any significant difference in the outcome measures between smokers and non-smoker, which showed no difference between them in sleep quality, physical activity or CRF.

With regards to the values of spirometry, apologies but these values were not available as in the software we have it is dimmed and there was no option to choose the value of spirometry such as FEV1 and FVC unless a software upgrade with a dongle was purchased from the supplier (which was not available at the time of data collection for the study).

Please add in table 1 the values of arterial blood pressure in maximal effort and at the 1st minute of recovery (HR, BP, VO2) to exclude the cardiological diseases. Several of participants are smoker and obese

Response

Thanks for this valuable comment. We did not aim to measure blood pressure throughout the test, but we kept an automated blood pressure machine to measure participants with any abnormal response to exercise testing. But we have measured blood pressure for all the participants immediately after the test during the recovery period. So we have added information about the HRR and blood pressure after 1 minute which was added into table 1. The absolute value of VO2 in (l/min) was added. 

Please refer in cause of each trial was terminated i.e. dyspnea, leg fatigue...etc

Response

Reasons for test termination was added to table 1 and the changes were highlighted in turquoise 

Please add in table 1 the ratio of VE/MVV

Response

Thanks for the comment. Unfortunately, this is one of the measures of spirometry which as we mentioned above is not available.

Please add the value of resting and maximal effort of PETCO2. The COSMED Quark record the end-tidal volume of CO2. The increased values of CO2 in resting relate to sleep deprivation.

Response

Many thanks to the reviewer for highlighting this valuable information about the end-tidal volume and the references provided. It really opened up ideas for future research. Resting PetCO2 and peak PetCO2 were added to table 1. It is a very good idea to examine sleep deprivation, which we think would show an inverse relationship as the reviewer and the published literature suggest. But we also believe that sleep deprivation should be measured objectively like previous research. As in the current study, non of the PSQI components were associated with Resting PetCO2 and peak PetCO2 (no data reported in the study). 

Reviewer 3 Report

Major

  1. smoke have a significant effect on CRF, which may result in bias of the analysis.
  2. Due to the relatively small sample size of 33 in the present study, more sound reason needs to provide for the sample size calculation. What did the authors mean by using "a from first 10 observations per the CRF (represented by the 74 VO2 peak values) and sleep quality as reported on the Pittsburgh sleep quality index ."?  Actually, the available published paper in the related field commonly had much more participants.

3. PA level was assessed based on their level of physical activity in the last seven days before testing. I would like to suggest take several times of the evaluation to avoid the possible accidental error.

Minor

some flaws in the format, as well as inappropriate table format

Author Response

We would like to thank the reviewer for the constructive feedback and comments. Responses and changes to the reviewer are highlighted in light green in the main manuscript

Smoke have a significant effect on CRF, which may result in bias of the analysis.

Response

Thanks for the comment. Yes, we agree that smoking has a significant effect on CRF. To eliminate any bias, an investigation of the relationship between CRF and smoking was conducted using point biserial correlation, which showed no significant association. This statement was added to the manuscript and highlighted in light green. Furthermore, we compared the results between smokers and non-smokers for the main outcome measures, which also showed no significant differences in any of the outcomes measures (CRF, PSQI, IPAQ).  

Due to the relatively small sample size of 33 in the present study, more sound reason needs to provide for the sample size calculation. What did the authors mean by using "a from first 10 observations per the CRF (represented by the 74 VO2 peak values) and sleep quality as reported on the Pittsburgh sleep quality index ."?  Actually, the available published paper in the related field commonly had much more participants.

Response

Thank you for this valuable comment. We agree with the reviewer that a sample size of 33 participants is relatively small. However, this sample size assures an adequate power to detect the relationship between VO2peak and sleep quality. We analyzed data from the first 10 participants enrolled in the study to determine the minimum sample size required to get meaningful results. Their data were used to measure the degree of association of VO2peak and sleep quality through Spearman's rank correlation [estimated sample correlation coefficient rs = 0.45 and the 95% confidence interval for the correlation were 0.102 and 0.700]. Based on these assumptions, we needed a sample size of 32 subjects to produce a two-sided 95% confidence interval with an actual width equal to 0.597. Here is the reference on which we based our decision in sample size calculation 

Bonett, D. G. and Wright, T. A. 2000. 'Sample Size Requirements for Estimating Pearson, Kendall and Spearman Correlations.' Psychometrika, Vol 65, No 1 (March), 23-28.

Looney, S. W. 1996. 'Sample size determination for correlation coefficient inference: Practical problems and practical solutions.' American Statistical Association 1996 Proceedings of the Section on Statistical Education, 240-245.

PA level was assessed based on their level of physical activity in the last seven days before testing. I would like to suggest take several times of the evaluation to avoid the possible accidental error.

Response

Thanks to the reviewers for raising this point. It was a very critical point. Indeed after reviewing the results of the IPAQ and re-entered them again into the excel sheet, we discovered that there was some accidental error in some of the data of the participants. Thanks for bringing attention to this point. we repeated the correlational analysis, and we found that there is an inverse significant association between IPAQ and PSQI total score. Major amendments were made to the results especially in figure 2 and table 3 and table 4, as well as the reported results. All the changes are highlighted in light green. Also, the amendment was made to the Abstract, results, and discussion. 

some flaws in the format, as well as inappropriate table format

Response

The format has been corrected and table 1 has been adjusted. 

Reviewer 4 Report

Lines 39-40: CRF is not used interchangeably with physical fitness. Please revise statement.

Line 80: “atthe” should be “at the”

Line 97: Why use ETT instead of graded exercise test?

Line 101- 102: delete sentence “the test…previous study” prevents redundancy

Lines 124-125: What is the rationale for analyzing the gas every 5 seconds instead of every 15 seconds?

Lines 126-127: What is the rationale for choosing peak over max? Why consider peak when most studies report max for treadmill studies? Peak is typically used for cycle ergometry. There needs to be discussion about this within the paper. 

Line 127: Formatting issue.

Line 134: Typo “anda” should be “and a”

Line 160: Why was the comparison data of smokers and non-smokers omitted from the study?

Line 164: Need to reference the flow diagram in the previous paragraph.

Line 215: VO2Peak/ VO2MAX indicates these terms are interchangeable when in fact they are distinctly different. Please revise

Line 238: Typo. “peopleas”

Line 241: What were similar outcomes measures that were used in the current study compared the ones mentioned? If there was a difference in methodology, that should be highlighted here.

Line 252-253: The authors state “people with low PA” it should be “people who reported low PA” since the IPAQ was used.

Lines 267-268: What units were used for the cost of energy? METs, kcals? This clarification along with a simple explanation as to why specific units were chosen will give more clarification to the readers.

Lines 287-297: There needs to be mentioning of subjective PA via IPAQ instead of objective PA through accelerometry.

What criteria was used to determine true VO2 peak? Were RER> 1.1, HR within Max HR, VO2 platue, verification bouts, etc.?

Throughout document use VO2PEAK and VO2MAX.

Author Response

Thanks and gratitude for the valuable comments and constructive feedback from the reviewer.  The changes in response to the reviewer are highlighted in grey in the new manuscript. 

Lines 39-40: CRF is not used interchangeably with physical fitness. Please revise statement.

Response

The statement has been revised and deleted in Lines 39-40

Line 80: “atthe” should be “at the”

Response

Apologies for these typos, which was made when paragraphs were transferred to the form recommended by the journal. This typo has been corrected. 

Line 97: Why use ETT instead of graded exercise test?

Response

Thanks for the comment. A graded exercise test was used, as we agree with the reviewer it can be more appropriate. It was changed throughout the test. 

Line 101- 102: delete sentence “the test…previous study” prevents redundancy

Response

The statement was deleted, as we felt we already described what we did with the protocol. 

Lines 124-125: What is the rationale for analyzing the gas every 5 seconds instead of every 15 seconds?

Response

Thanks for this comment. We used 5 seconds as this is the minimum sampling rate provided from The COSMED Quark CPET. if this was not used, the sampling would be random. For example in 1 minute (we would have readings of VO2 3sec, 5sec. 6sec, 3sec, 4sec). Therefore, we have chosen 5 seconds to be the standardized averaged VO2. Then, to minimize fluctuations and artefacts the final collected data would be presented every 30 seconds interval (taking the average of six VO2 readings (6 VO2 reading x 5 =30 seconds interval of VO2) and because we used incremental increase every 1 minute, we chosen 30 seconds interval for the final analysis. The recommended sampling rate is 20-30 seconds 

Mezzani, A. (2017). Cardiopulmonary Exercise Testing: Basics of Methodology and Measurements. Annals of the American Thoracic Society. 

Lines 126-127: What is the rationale for choosing peak over max? Why consider peak when most studies report max for treadmill studies? Peak is typically used for cycle ergometry. There needs to be discussion about this within the paper. 

Response

Thanks for this comment. Indeed, most of the studies utilized VO2 max on treadmill. However, one of the most important criteria to successfully obtain VO2 max is the attainment of a plateau in oxygen uptake over a period of time, whereas, VO2 peak indicate the highest maximum attainment of oxygen uptake without necessarily reaching a maximal effort. All of the participants included in the analysis failed to achieve a plateau in oxygen uptake over a period of time. Thus, VO2 peak was utilized. Furthermore, a paragraph was added to the discussion and highlighted in yellow as this was also raised by one of the reviewers lines 289-307.  

Line 127: Formatting issue

Response

Thank you. Now it is corrected.

Line 134: Typo “anda” should be “and a”

Response

Thanks. Now it is corrected

Line 160: Why was the comparison data of smokers and non-smokers omitted from the study?

Response

Good point. This was omitted from the study because it is beyond its scope. We did not aim to compare smokers and non-smokers, but we ran the analysis to exclude any effect of smoking on the data. But we think because it was mentioned by more than one reviewer that it is a critical point to be reported. Now in the new version, it is reported and we also reported the correlation between CRF and smoking in the current sample. The changes are highlighted in grey.

Line 164: Need to reference the flow diagram in the previous paragraph

Response

The flow diagram has been referenced in the previous paragraph. 

Line 215: VO2Peak/ VO2MAX indicates these terms are interchangeable when in fact they are distinctly different. Please revise

Response

Thanks. The statement has been revised line 243

Line 238: Typo. “peopleas

Response

Thanks for highlighting this. Corrected. line 266

Line 241: What were similar outcomes measures that were used in the current study compared the ones mentioned? If there was a difference in methodology, that should be highlighted here.

Response

This was addressed by adding a table that summarizes the similarities and differences between the current study and the previous ones. Table 5

Line 252-253: The authors state “people with low PA” it should be “people who reported low PA” since the IPAQ was used.

Response

Thanks for this. This was corrected in line 304

Lines 267-268: What units were used for the cost of energy? METs, kcals? This clarification along with a simple explanation as to why specific units were chosen will give more clarification to the readers.

Response

Thanks for this comment which is important to clarify to the readers. A statement was added in lines 320-322 which add clarification. 

Lines 287-297: There needs to be mentioning of subjective PA via IPAQ instead of objective PA through accelerometry.

Response

Thanks for this reminder. A statement was added in the limitation in lines 350-353. 

What criteria was used to determine true VO2 peak? Were RER> 1.1, HR within Max HR, VO2 plateau, verification bouts, etc.?

Response

Thanks for raising this point. We were aiming to exercise test the participants to the maximum in order to get the Vo2 max. However, none of the participants achieved plateue of VO2 increase. Thus, we recorded peak VO2 uptake and RER 1.1 was the main criterion to make sure that the participants at least exercised to their maximum ability. A statement was added to lines 161-166. 

Throughout document use VO2PEAK and VO2MAX. 

Response

The manuscript amended according to the requested changes. Thanks.

Round 2

Reviewer 1 Report

Appropriate changes were made to the manuscript in accordance with the reviewer's recommendations.However, in my opinion, they still haven't presented the reasons for such a low CRF level in young healthy people. This is a significant limitation of these studies.

Author Response

Appropriate changes were made to the manuscript in accordance with the reviewer's recommendations. However, in my opinion, they still haven't presented the reasons for such a low CRF level in young healthy people. This is a significant limitation of these studies.

Response

Many thanks for the comments and suggestion for the manuscript. With regard to the reasons why our sample had low CRF, this can be due to BMI of the participants. As most of them were above normal weight. An investigation was added in the results showing that there was an inverse association between CRF and BMI, but had no influence on the association between CRF and PSQI. The new changes are highlighted in Yellow. A paragraph was added in the discussion part to discuss what was mentioned above. 

Reviewer 2 Report

Authors spend many efforts to improve the manuscript according to the review suggestions. I thank the authors for addressing my concerns.

Author Response

Authors spend many efforts to improve the manuscript according to the review suggestions. I thank the authors for addressing my concerns.

Response

Many thanks to the reviewer for highlighting very important points in the manuscript and to be considered in future research. 

Reviewer 4 Report

The authors have taken the time to address all of the suggestions. These suggestions have strengthened the paper that will add to the body literature. 

Just a couple of minor suggestions:

Line 15: Physical activities should be physical activity

Table 1: Did authors follow WHO guidelines for BMI classification? If so, pre-obesity should be overweight.

Table 6: This is a good addition, but it should be formatted to fit the margins of the paper.

Author Response

The authors have taken the time to address all of the suggestions. These suggestions have strengthened the paper that will add to the body literature.

Response

Thanks for raising these suggestions which indeed strengthened the paper. All the changes in the manuscript according to the suggestion from the reviewer are highlighted in light green. 

Line 15: Physical activities should be physical activity

Response

Many thanks for pointing this. Amended. line 15 and also highlighted in light green. 

Table 1: Did authors follow WHO guidelines for BMI classification? If so, pre-obesity should be overweight.

Response

Yes, it was based on WHO guidelines, and it has various classification from different sources. But yes we agree it is better to be classified as overweight. This was changed in table 1. and line 198. 

Table 6: This is a good addition, but it should be formatted to fit the margins of the paper.

Response

Thank you for raising this point about discussing other similar studies. We did our best to change the size of the table not to cross the margins of the paper. We hope it is convenient now.